# multicrispr: gRNA design for prime editing and parallel targeting of thousands of targets

Aditya M Bhagwat , Johannes Graumann, Rene Wiegandt , Mette Bentsen, Jordan Welker , Carsten Kuenne , Jens Preussner , Thomas Braun, Mario Looso

Targeting the coding genome to introduce nucleotide deletions/insertions via the CRISPR/Cas9 technology has become a standard procedure. It has quickly spawned a multitude of methods such as prime editing, APEX proximity labeling, or homology directed repair, for which supporting bioinformatics tools are, however, lagging behind. New CRISPR/Cas9 applications often require specific gRNA design functionality, and a generic tool is critically missing. Here, we introduce multicrispr, an R/bioconductor tool, intended to design individual gRNAs and complex gRNA libraries. The package is easy to use; detects, scores, and filters gRNAs on both efficiency and specificity; visualizes and aggregates results per target or CRISPR/Cas9 sequence; and finally returns both genomic ranges and sequences of gRNAs. To be generic, multicrispr defines and implements a genomic arithmetic framework as a basis for facile adaptation to techniques recently introduced such as prime editing or yet to arise. Its performance and design concepts such as target set–specific filtering render multicrispr a tool of choice when dealing with screening-like approaches.

## Introduction

CRISPR loci, first reported in 1987 (Ishino et al, 1987), later realized to constitute a prokaryotic immune system (Bolotin et al, 2005; Mojica et al, 2005; Pourcel et al, 2005), have now been transformed into a versatile molecular tool kit for genome engineering and analysis (Gasiunas et al, 2012; Jinek et al, 2012; Cong et al, 2013). Molecularly, the system comprises two components (Fig 1A): the Cas9 enzyme, which introduces double stranded cuts into DNA, and a gRNA. The latter consists of a *scaffold* linked to a 20-nucleotide *spacer* (N20, N = A, C, G, or T). When Cas9 binds to an NGG motif (*protospacer adjacent motif*, PAM), DNA is cleaved if the sequence immediately upstream matches the spacer.

The CRISPR/Cas9 application portfolio is constantly growing at considerable speed. One notable recent innovation is prime editing (Anzalone et al, 2019) (Fig 1B). In its initial form (named PE2), this technology fuses a *Cas9 nickase* (cutting a DNA on a single strand) to a *reverse transcriptase* and combines it with an extended gRNA consisting of a spacer, a scaffold, a *primer binding site*, and a *reverse transcription template* (named as 3' extension). While the *spacer* continues to guide the complex to a genomic locus, the primer binding site also binds to a region in the target DNA on the PAM strand that serves as a primer for reverse transcription using the additionally provided reverse transcription template (Fig 1B). This prime editor allows rewriting of up to 48 nucleotides at a specific locus of interest, enabling knockout, knock in, and precision editing. In a more recent development named as the PE3 system, an additional nicking spacer was added, able to perform a single stranded cut on the opposite strand about 40–90 nucleotides downstream of the prime editing site. This modification increases editing efficiency but is also associated with a higher probability of indel events.

A further example for a recently emerged CRISPR/Cas9 application is parallel targeting of many loci with gRNA libraries, required for instance when targeting transcription factor binding sites (TFBSs) or their neighborhoods (Shariati et al, 2019). Other screening-oriented CRISPR/Cas9-based applications include genome-wide visualization (Zhou et al, 2017) or complex gRNA libraries to investigate cell fitness (Wegner et al, 2019). For such applications, the total number of gRNAs, or library complexity, directly correlates with effort and costs.

In general, several N20-NGG CRISPR sites may be identified for a genomic target region, but not all of them are equally suited. Thus, gRNA design, defined as the process of finding a good gRNA (set), involves two major tasks beyond the identification of N20NGG sequences within the target region: 1) off-target analysis to select spacers with a minimum number of (mis)matches to other genomic positions and 2) on-target scoring to select spacers expected to target the region of interest efficiently (using sequence-based prediction models).

Parallel targeting in a genome-wide context implies additional gRNA design needs. For one, the number of simultaneously targeted sequences may be large and processing efficiency thus is essential. In addition, for example, when targeting TFBSs, these target sequences are also prone to be very similar, as they conform to a

Max Planck Institute for Heart and Lung Research, Bad Nauheim, Germany

Correspondence: mario.looso@mpi-bn.mpg.de

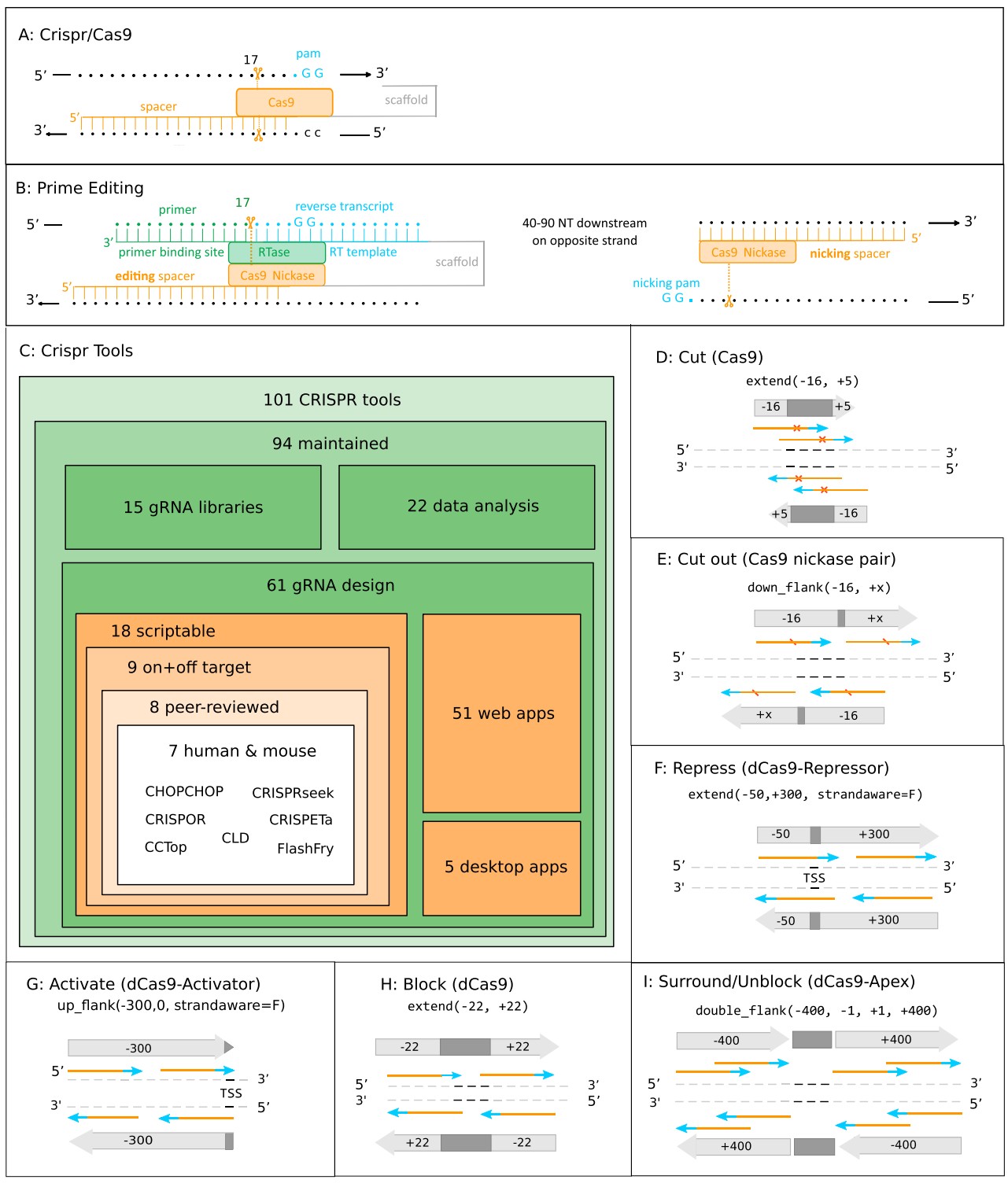

**Figure 1. Schematic representation of CRISPR/Cas9 application and arithmetic.**
**(A, B)** illustrate the basic mechanism of CRISPR/Cas9 and prime editing. Both systems target a genomic region based on complementarity to a 20-nucleotide spacer sequence (when followed by NGG on the opposite strand), and both involve cutting the PAM–strand spacer after position 17 (double or single strand). **(B)** The prime editor (B) additionally enables editing of the sequence following nucleotide 17 through reverse transcription of a template (light blue, provided as a gRNA component), a process which is initiated through pairing of the primer binding site (another gRNA component) with the primer (a portion of the spacer on the PAM–strand). **(C)** A graphical overview of existing CRISPR/Cas9 gRNA design tools as provided by Torres-Perez et al (2019) and their classification. **(D, E, F, G, H, I)** genomic arithmetic as needed for

consensus motif that can occur thousands of times. Consequently, CRISPR/Cas9 spacers often match multiple targets. While this is undesirable for traditional CRISPR-based techniques targeting single loci, it can be used for parallel targeting, allowing for a smaller gRNA set. The idea of parallel targeting thus requires differentiating between (mis)matches within the target set and to the genome, a process we define as target set–specific filtering (TSSF).

To rise to the challenge of gRNA design, many software tools have been developed in recent years. As summarized in Fig 1C, the "WeReview:CRISPR web table" (Torres-Perez et al, 2019) reports as many as 101 tools supporting CRISPR-based technology. Narrowing this down to currently available scriptable gRNA tools that perform both off-target analysis and on-target scoring, which are additionally peer-reviewed and support at least human and mouse as target organisms, this number reduces to only seven. Ordered by number of citations, these are CHOPCHOP ([Montague et al, 2014; Labun et al, 2016; Labun et al, 2019], 1,204 citations), CRISPOR ([Haeussler et al, 2016; Concordet & Haeussler, 2018], 706 citations), CCTop ([Stemmer et al, 2015], 402 citations), CRISPRseek ([Zhu et al, 2014], 131 citations), CLD ([Heigwer et al, 2016], 44 citations), CRISPETa ([Pulido-Quetglas et al, 2017], 41 citations), and FlashFry ([McKenna & Shendure, 2018], 21 citations) (citations were retrieved from Google Scholar on 29 July, 2020). As of August 2020, however, these tools lack functionality to support more recently introduced applications of CRISPR/Cas9. For instance, prime editing is currently supported by CRISPRseek only. Others are suffering from performance constraints (see the Results/Benchmarking section).

From a more general viewpoint, a future-proof CRISPR/Cas9 design tool requires the ability to keep up with the fast pace of new developments in this field. This demands an ongoing update of the implementation, accompanied by a continuous integration circuit and timely releases of new versions with updated functionality (Hanna & Doench, 2020). This in turn requires a generalized framework and coding paradigm which allows for an efficient extension and adaptation to new applications while maintaining backward compatibility. In this context, we propose generic genomic arithmetic as an essential feature to render a gRNA design tool suitable for the plethora of already established, as well as future, CRISPR-based strategies (Fig 1D–I). Cutting within a target site, for instance (Fig 1D), requires a strand-specific [−16 +5] extension before searching for N20-NGG spacer–PAM sequences, to ensure that Cas9, which has a cut site after nucleotide 17, cleaves within the target range. Excising a target site (Fig 1E) with a Cas9/nickase pair, and possibly fixing it with homology directed repair, require downstream flanking by [−16, +x] to find a Cas9/nickase pair in "PAM-out orientation" for excision (a Cas9/nickase pair in the [−x, +5] upstream flanks with a "PAM-in" orientation has been experimentally shown to be ineffective [Gearing, 2018]). Repressing gene expression via a dCas9-repressor approach (Fig 1F), on the other hand, requires a [−50, +300] extension of the TSS before spacer–PAM sequence search

(Gilbert et al, 2014), an operation carried out without consideration of the strand. By contrast, when activating a gene with a dCas9-activator approach (Fig 1G), a [−300, 0] strand-agnostic (Doench, 2020) upstream left-flanking of the transcription start site is required before spacer/PAM sequence search, allowing activator binding in relevant promoter/enhancer regions. Finally, blocking a target site (Fig 1H) requires a [−22, +22] target extension before spacer/PAM search, ensuring that at least one nucleotide of the target area gets blocked. As indicated in (Fig 1I), targeting the vicinity of a sequence requires searching for CRISPR sequences in both flanks around the target site. The latter is, for example, needed for nucleo–protein complex purification via an affinity-tagged dCas9 (Liu et al, 2017), as well as Apex2-based protein interactor biotinylation (Myers et al, 2018).

A subset of the existing software tools offer limited genome arithmetic functionality, such as options to specify a TSS offset between a target sequence and CRISPR spacer/PAM sequences (e.g., CHOPCHOP and CRISPRseek). However, the increasing variety of modern CRISPR applications renders the flexibility resulting from a generic genome arithmetic framework indispensable. Summarizing, an easily extensible CRISPR design tool able to encompass both, current and future CRISPR methods, will profit from a comprehensive genome arithmetic vocabulary that does not yet exist in this form.

# Results

## The multicrispr package

Motivated by the need for a gRNA design tool with generic functionality supporting a multitude of CRISPR approaches, we developed the R package multicrispr. It has been designed to be highly performant and user-friendly and provides a comprehensive genome arithmetic vocabulary.

As outlined in Fig 2A, a typical multicrispr workflow consists of five sequential steps. Each step provides optional plotting functionality, as exemplified for two exemplary applications in the next section. To integrate seamlessly into the R/Bioconductor environment, a GRanges object (a core Bioconductor class) is returned as a final result, including information on both off- and on-target analyses. In addition, multicrispr generates human-readable and machine-parsable tab separated value files for further downstream processing.

The workflow starts with defining targets for genome engineering by providing either genomic coordinates or genomic identifiers. These targets are transformed into spacer/PAM targets through *extension* and/or *flanking* (upstream and downstream) operations as required for the individual CRISPR application performed (discussed earlier and in Fig 1). In the subsequent step, the transformed target ranges are searched for spacer/PAM sequences. By default the wild-type *Streptococcus pyogenes* N20-NGG (spacer/PAM)

individual CRISPR/Cas9 applications as indicated. Black lines represent the target range, orange arrows indicate the spacer sequences, blue arrows are PAM sequences, orange crosses depict Cas9 cut sites, and large arrows mark the search region for spacer–PAM sequences.

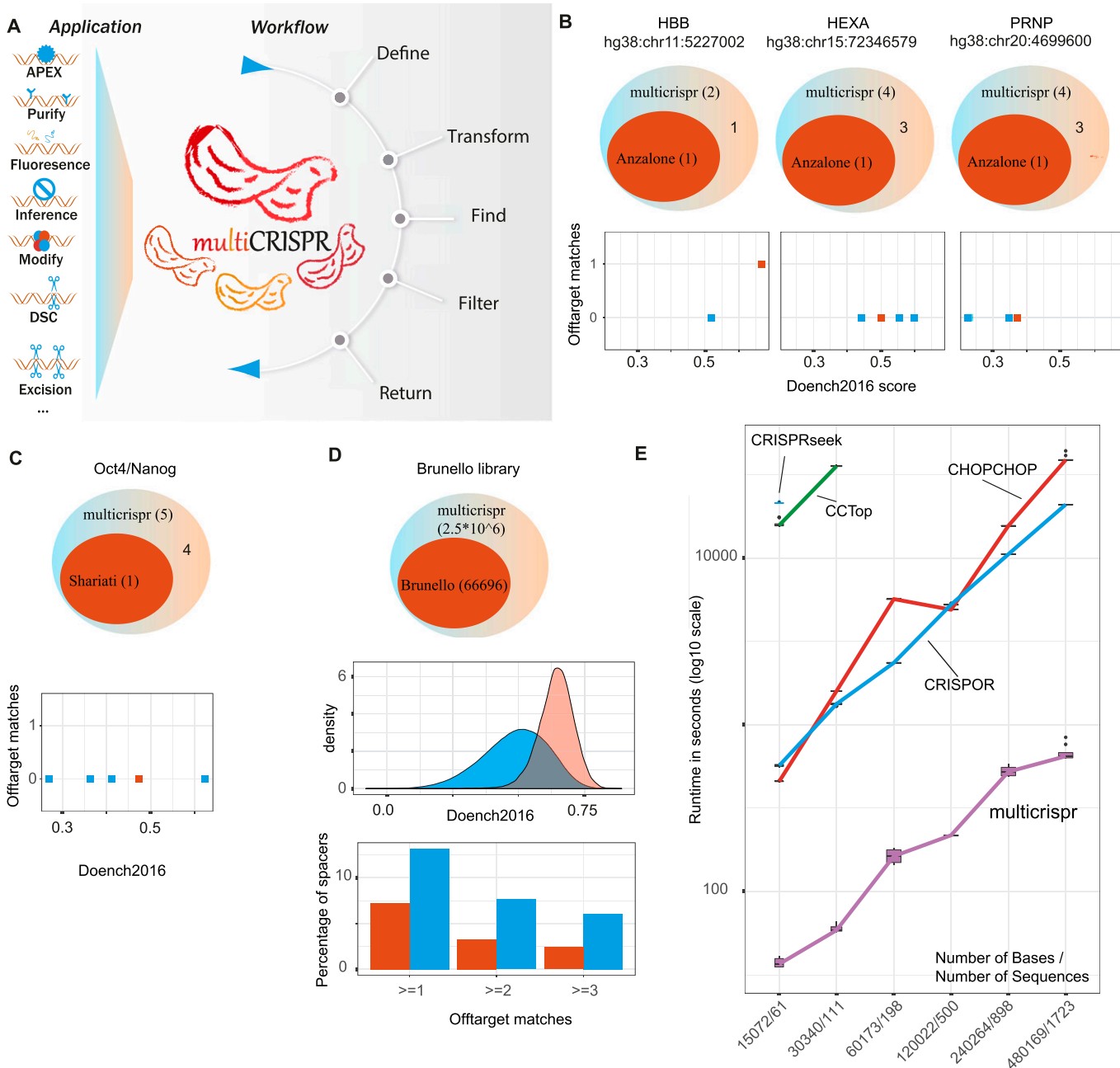

**Figure 2. multicrispr workflow and validation.**
**(A)** Selection of supported CRISPR applications and workflow of multicrispr. **(B)** Overlap of prime editing spacer output of multicrispr and spacers used for the sickle cell locus in the *HBB* gene, the Tay–Sachs locus in the *HEXA* gene, and the prion disease locus in the *PRNP* gene, as given by Anzalone et al (2019). Scatter plots indicate scores and #mismatches given for all spacers found by multicrispr for the respective loci. **(C)** Overlap of multicrispr spacers and spacers used to block Oct4 TFBS [−151, −137] upstream of the Nanog gene, as used in Shariati et al (2019). Scatter plots indicate scores and #mismatches given for all spacers found by multicrispr for the respective loci. **(D)** Overlap of spacers identified with multicrispr for all Brunello exons (Doench et al, 2016). Density plot indicates scores for spacers specific for multicrispr (blue) and overlapping Brunello (red). Bar plots indicate # mismatches for these spacer sets as well. **(E)** Runtime comparison of gRNA design tools: the x-axis depicts the increasing number of input sequences and total bases, respectively, whereas the y-axis shows the total time needed by individual tools to design respective gRNAs on a $\log_{10}$ scale in seconds. Colors represent individual tools. Box plots represent repetitive processing of each input file (n = 10) to control for variability in computing performance.

sequence is sought, but alternative spacer/PAM sequences may be specified as well. In the next step, filtering of identified gRNAs is performed based on off-target and on-target parameters. For each identified spacer, the number of off-target (mis)matches is reported.

The number of allowed mismatches in off-target filtering varies according to the CRISPR application conducted. In the case of parallel targeting, exact matches with a variable amount of mismatches are allowed, and cross-target matches are considered in on-target effects,

as discussed earlier (TSSF). For prime editing, which has been reported to be mismatch-free (Anzalone et al, 2019), only exact matches are considered and cross-target matches are rejected. On-target scores per gRNA are added in a final step of filtering, providing extra guidance on which spacers to select for the experiment. multicrispr supports the "Doench2016" (Doench et al, 2016) as well as the "Doench2014" (Doench et al, 2014) scoring models, the first of which is the current gold standard for gRNA efficiency prediction (Haeussler et al, 2016). Finally, identified spacer ranges and sequences, together with off-target counts and on-target scores, are returned as a GRanges object. Of note, multicrispr also provides functions for writing a GRanges object to and reading it from a txt file.

### Validation

To validate the spacers identified by multicrispr, we applied our tool to gold standard targets in three different CRISPR applications, with publications providing experimentally tested spacers of proven functionality. First, we used multicrispr to identify spacers for prime editing the sickle cell locus in the *HBB* gene, the Tay–Sachs disease locus in the *HEXA* gene, and the prion disease locus in the *PRNP* gene, each of which were successfully prime edited by Anzalone et al (2019). Using multicrispr and as illustrated in Fig 2B, we confirmed the spacers used by Anzalone et al (2019) for all loci. Of note, we were able to derive additional spacers targeting the same editing sites with scores and genomic mismatches comparable with the published controls.

In a second comparison, multicrispr was instructed to identify spacers blocking an Oct4 TFBS located upstream of the Nanog gene, reproducing work by Shariati et al (2019). In this case, multicrispr genome arithmetic functionality for upstream flanking was required to extend the target region by ±22 nucleotides, as discussed earlier and detailed in Fig 1D–I. Aside from verifying the published spacer, multicrispr identified four additional spacers (Fig 2C), one of which is characterized by a higher Doench2016 targeting efficiency score.

Finally, we used multicrispr to search for spacers in the exons targeted by the Brunello library (Doench et al, 2016), a validated gRNA library with 76,441 spacers targeting 19,114 transcripts (each transcript residing in a different gene, and each being targeted by up to four different gRNAs). After mapping to a current genome version and gene IDs (see the Materials and Methods section), 66,696 of these spacers overlapped an exon, subsequently named as *Brunello exons*. Using these as target ranges, multicrispr identified more than 2.5 million spacers in total, including all 66,696 Brunello spacers (Fig 2D top). For each of them, the Doench2016 score was computed and a genome-wide off-target analysis was performed. As expected, the Brunello spacers are characterized by an enrichment in high Doench2016 scores (Fig 2D center). Genome-wide off-target analysis revealed most identified spacers to be unique (Fig 2D bottom). multicrispr, having been designed to scale to large datasets, performed both operations (Doench2016 scoring and genome [mis] match analysis) for the 2.5 million spacers within 1.1 and 1.6 h, respectively, utilizing a 15 Core/128 GB RAM Linux virtual machine (see also Fig 2E and performance tests in the following text).

Taken together, these examples confirm multicrispr's ability to reproduce experimentally validated spacers efficiently for small- and large-scale screening–like applications.

### Benchmarking and feature comparison

From a practical point of view, performance of a computational tool is important, especially when it is not typically run on a horizontally scaling cluster, which is common for R analysis pipelines. To test the performance of multicrispr systematically, we performed comparative benchmarking with the four most popular tools, including spacer identification, Doench2016 on-target scoring, and off-target analysis in the benchmarking. These operations were performed on six increasingly larger target sequence sets using the same 11 Core/132 GB RAM Linux machine. The sequences in these target sets were identical per run across the tools, and an average of 10 runs was calculated per target set size (Fig 2E). For CRISPRseek and CCTop, computation time for the two smallest target sets was in the range of days to process, and these were thus excluded from further benchmarking. The remaining tools (CRISPOR, CHOPCHOP, and multicrispr) all met the challenge, with significant performance differences, however. Whereas CHOP-CHOP and CRISPOR needed processing time on the scale of hours, multicrispr finished the job within minutes (accelerating the search by factor 21 when compared with the second fastest tool, CRISPOR). As a result, the Brunello exon example mentioned earlier is out of scale for all tested tools, except multicrispr in terms of runtime.

We conclude that some of the performance differences are driven by the algorithms used for off-target analysis. Whereas CRISPRseek uses the Aho and Corasick (1975) algorithm (which is exact), CHOPCHOP, CRISPOR CCTop, and multicrispr use Bowtie1 (Langmead et al, 2009). However, moving from exact string matching (Aho–Corasick) to fast read mapping (Bowtie1) creates a potential for precision loss, and earlier publications exist, reporting, for example, Bowtie1 to miss off-targets (Doench et al, 2016). To better understand to which extent such precision loss occurs, we performed a benchmark in which we identify genome (mis)matches using both methods: the exact but slow Aho–Corasick algorithm and the fast read mapper Bowtie1 (both available in R as BSgenome:: vcountPDict function and RBowtie package). We took four of the main prime editing targets of Anzalone et al (2019) and searched for spacers (using multicrispr's default parameters). For the 10 identified prime spacers (nine of which also accompanied by a nicking spacer [black] Fig 3A), we then performed a (PAM-agnostic) genome (mis)match analysis using both methods (Aho–Corasick and Bowtie1). The results are shown in Table 1. Bowtie achieves perfect precision when it comes to exact-match and single-mismatch off-targets. However, for double mismatches, some off-targets are missed, and for triple mismatches, a larger number of off-targets are missed. Unfortunately, the Aho–Corasick algorithm is prohibitively slow for large- and even medium-scale applications. For the illustrated set of 10 spacers, which Bowtie1 executed within a second, Aho–Corasick took 95 min to complete. Large-scale parallel targeting applications with the Aho–Corasick algorithm are, therefore, not realistic. We conclude to use Bowtie1 for large- and medium-scale applications, and when runtime is not an issue, Aho–Corasick may be preferable. To allow for this, multicrispr allows switching between the two methods.

One question that remains, though, is why the performance of multicrispr so strongly exceeds that of the other tools, which in

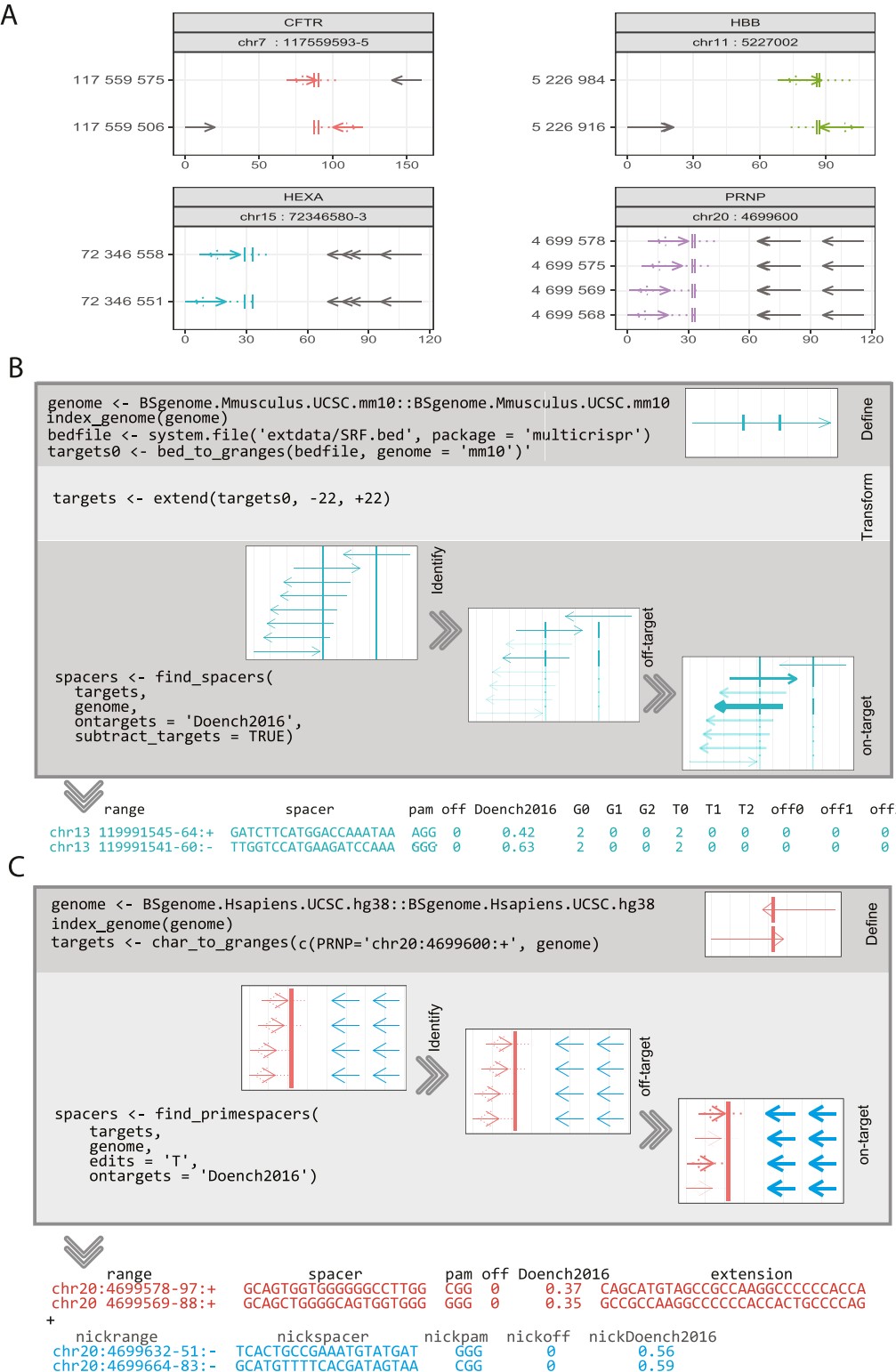

**Figure 3. Prime editing spacers and use cases of multicrispr.**
**(A)** Off-target benchmarking was performed using 10 prime editing spacers (colored solid lines) to target the four main prime editing loci of Anzalone et al (2019), (colored vertical bars); the cystic fibrosis locus in the *CFTR* gene (red), the sickle cell anemia locus in the *HBB* gene (green), the Tay–Sachs disease locus in the *HEXA* gene (blue), and the prion disease locus in the *PRNP* gene (purple). Nicking spacers are shown with black lines for completeness but were not used for off-target benchmarking. Genomic coordinates are shown on the y-axis, and additional offsets are shown on the x-axis. **(B)** The parallel targeting of 1,974 binding sites of the transcription factor SRF. Boxes show results for one particular binding site (chr13:119991554-69:+), indicating the genomic locus on y-axis and range width on x. multicrispr finds eight spacers for this binding site. Three of them are target-specific (nonspecific spacers are faded out). Two of them are predicted to have a good targeting efficiency (Doench2016 is mapped to line thickness). The resulting GRanges object is presented as a table (T, target [mis]match counts; G, genome [mis]match counts; off, off-target counts, number 0–2 indicates number of mismatches). **(C)** Prime editing the prion disease locus in the *PRNP* gene. Primer binding site and reverse transcription template, jointly referred to as 3′ extension, are shown with dotted lines.

principle use the same off-target analysis algorithm. The main difference is not in the used aligner, rather the way that multi-target sets are handled. Multicrispr has large target sets at the core of its design since inception, performs a single joint Bowtie run for all targets in the set, and then later, aggregates the results. The other three tools all perform off-target analysis by running separate Bowtie instances for multiple targets, an approach that does not scale toward sets with a large number of targets.

**Table 1. Number of hg38 genome matches, considering 0–3 mismatches, using two different methods (A, Aho–Corasick; B, Bowtie).**

| Target | CRISPR site | | | No. of mismatches | | | | | | | |
|---|---|---|---|---|---|---|---|---|---|---|---|
| | | | | 0 | | 1 | | 2 | | 3 | |
| Gene | Spacer range | Spacer sequence | PAM | A0 | B0 | A1 | B1 | A2 | B2 | A3 | B3 |
| CFTR | chr7:117559575-94:+ | ATTAAAGAAAATATCATCTT | TGG | 1 | 1 | 7 | 7 | 145 | 143 | 2,304 | 2,277 |
| | chr7:117559606-25:– | TCTGTATCTATATTCATCAT | AGG | 1 | 1 | 7 | 7 | 125 | 124 | 1701 | 1,672 |
| HBB | chr11:5227003-22:– | CATGGTGCATCTGACTCCTG | AGG | 2 | 2 | 0 | 0 | 14 | 13 | 210 | 208 |
| | chr11:5226984-7003:+ | GTAACGGCAGACTTCTCCTC | AGG | 1 | 1 | 0 | 0 | 7 | 7 | 83 | 82 |
| HEXA | chr15:72346551-70:+ | TGTAGAAATCCTTCCAGTCA | GGG | 1 | 1 | 0 | 0 | 25 | 25 | 295 | 292 |
| | chr15:72346558-77:+ | ATCCTTCCAGTCAGGGCCAT | AGG | 1 | 1 | 0 | 0 | 6 | 6 | 201 | 199 |
| PRNP | chr20:4699568-87:+ | AGCAGCTGGGGCAGTGGTGG | GGG | 1 | 1 | 2 | 2 | 95 | 88 | 904 | 891 |
| | chr20:4699569-88:+ | GCAGCTGGGGCAGTGGTGGG | GGG | 1 | 1 | 12 | 12 | 98 | 93 | 1,043 | 1,024 |
| | chr20:4699575-94:+ | GGGGCAGTGGTGGGGGGCCT | TGG | 1 | 1 | 2 | 2 | 55 | 54 | 857 | 826 |
| | chr20:4699578-97:+ | GCAGTGGTGGGGGGCCTTGG | CGG | 1 | 1 | 0 | 0 | 32 | 31 | 417 | 412 |

To benchmark multicrispr against a tool that does scale to large-scale target sets, we next performed a comparison to FlashFry (McKenna & Shendure, 2018) with 2,700 input sequences running on the same Linux machine that was used for the other benchmarks. It was not included in the first benchmark because it lacks support for Doench2016 (instead performing on-target scoring using the outdated Doench2014 which is supported by multicrispr as well). The results, summarized in Fig S1, confirm FlashFry's high-performance characteristics. Of note, multicrispr performs in the same order of magnitude.

**Table 2. Feature comparison of gRNA design tools.**

| | multicrispr | CHOPCHOP | CRISPOR | CCTop | CRISPRseek | FlashFry |
|---|---|---|---|---|---|---|
| | R | Py | Py | Py | R | Scala |
| (1) Install | | | | | | |
| One-liner | ✓ | | | | ✓ | ✓ |
| (2) Define targets | | | | | | |
| Target range(s) | ✓ | ✓ | ✓ | | | |
| Target gene(s) | ✓ | ✓ | | | | |
| Target sequence(s) | | ✓ | ✓ | ✓ | ✓ | ✓ |
| (3) Transform targets | | | | | | |
| Genome arithmetic | ✓ | | | | | |
| (4) Find spacers | | | | | | |
| Spacer sequences | ✓ | ✓ | ✓ | ✓ | ✓ | ✓ |
| Spacer ranges | ✓ | ✓ | | | | ✓ |
| Prime editing sequences | ✓ | | | | ✓ | |
| Prime editing ranges | ✓ | | | | | |
| (5) Count off-targets | | | | | | |
| Genome (mis)match algorithm | Aho+ Bowtie | Bowtie | Bowtie | Bowtie | Aho | FlashFry |
| Genome (mis)match aggregation | ✓ | ✓ | ✓ | | | |
| Target cross-(mis)match subtraction | ✓ | | | | | |
| (6) Score on-targets | | | | | | |
| Doench2016 | ✓ | ✓ | ✓ | | ✓ | |
| Doench2014 | ✓ | ✓ | ✓ | | ✓ | ✓ |
| Labuhn 2017 (Labuhn et al, 2017) | | | | ✓ | | |

Next, we assessed the functionality of given gRNA design tools. Table 2 shows a detailed side-by-side comparison from the perspective of implemented features. In brief outline, multicrispr and CRISPRseek are both easy to install, whereas the others require the installation of long lists of dependencies. In terms of target definition, multicrispr processes targets defined as genomic ranges or gene names, instead of sequences. This allows reporting the identified spacers in relation to their original target ranges and facilitates additional upstream or downstream analysis steps, such as the application of various BED format–based tools. Although target range definitions are possible for CRISPOR and CHOPCHOP, the latter only accepts a single target per run. Explicit and extensive genome arithmetic functionality as defined in the background section is exclusively provided by multicrispr, with only limited implicit functionality provided by the other tools. Prime editing spacers (which imply additional sequence constraints) are found by only multicrispr and CRISPRseek. Off-target analysis is performed by all programs; however, the functionality provided is limited. Whereas CHOPCHOP and CRISPOR also aggregate genome match counts, only multicrispr implements TSSF, which additionally requires target cross-(mis)matches to be investigated. In comparison to CRISPRseek, the only other R package in the set, multicrispr facilitates the access to the Doench2016 scoring, by building on the reticulate framework (Ushey et al, 2020) and allowing for within-R installation and single-line use of the python module azimuth from the Doench laboratory directly.

In summary, we found multicrispr to lead other tools with respect to universality of application and extensibility, as well as smoothness of integration into its programming environment. Multicrispr thus defines a new standard with respect to performance and applicability in the context of large-scale gRNA library design.

### Use case 1: parallel targeting of SRF binding sites

In a first example, we use multicrispr's functionality to design gRNAs to block the 1,974 binding sites of the mouse transcription factor SRF by a CRISPRi approach (Qi et al, 2013). These binding sites are 16 nucleotides wide on average and form small variations of the SRF consensus motif (MA0083.3 from JASPAR database [Fornes et al, 2020]). As shown in Fig 3B, at first, indexing of the mouse genome is performed. This is a time-consuming operation, but it needs execution only once for any genome. To reduce execution time, we provide pre-built indices for the most common genomes at our S3 storage, and by default, these are downloaded. Next, the ranges are first subjected to a [−22, +22] flank extension to ensure that at least one spacer nucleotide is within the target range. multicrispr subsequently derives strand-specific spacer/PAM sites for the given targets. Plots (which are generated by default) give an intuitive visualization of the overall effects of each subsequent operation. Next, TSSF is performed, excluding sequences with off-target (mis)matches, when they are outside the target set. This is an essential step when dealing with large numbers of highly similar target ranges such as TF binding sites. For instance, the shown spacer sequence GTGAGAAGGTCGCCTTTATT has no genome or target mismatches (G1 = G2 = T1 = T2 = 0), but it does have two genome (perfect) matches (G0 = 2), each at different locations in the genome: one is the spacer range itself (chr13:119991560-79:−) and

another further downstream on the same chromosome (chr13: 120070949-68:+, not shown in the Figure). Interestingly, this other locus is also among the initially given targets, which means two target ranges are hit by this spacer (T0 = 2), bringing the number of off-targets down to zero (off0 = G0-T0 = 2-2 = 0). Because no further mismatches are given, this spacer is off-target–free (off = off0 + off1 + off2 = 0). Finally, multicrispr scoring functionality selects spacers with high efficiency potential, before a GRanges object including information on spacer (and target) ranges, spacer (and PAM) sequences, on-target scores (Doench2016), and target and genome (mis)matches is returned.

### Use case 2: prime editing of the prion disease locus

In the second example, we design gRNAs for prime editing the prion disease locus, a single nucleotide variation at chr20:4699600:+ (G→T) which confers resistance to prion disease and which was used as a showcase for prime editing by Anzalone et al (2019). gRNA design for prime editing is more complex, as was shown in Fig 1B. In a first step, the genome of interest, as well as the genomic location of the single nucleotide variation, is thus specified as input (Fig 3C).

Next, multicrispr finds prime editing spacers, performing a two-way [−5, +nrt+16] extension around the prime editing target (nrt = number of reverse transcript nucleotides). In their seminal prime editing publication, Anzalone et al (2019) suggest a default nrt value of 16, which is the value used here. For each prime editing spacer, 3' extensions and nicking spacers are identified as well. Off-target counting and on-target scoring are performed for both editing and nicking spacers, with off-target analysis considering only exact matches for editing spacer and mismatches for nicking spacers.

Thus, parameterized multicrispr identifies four prime editing spacers and two nicking spacers (Fig 3C). All spacers are found to be off-target free. Both nicking spacers have good Doench2016 scores, as do two of the four editing spacers (note, however, that for editing spacers, the relevance of Doench2016 scores still awaits further validation). Finally, the identified spacers are returned as a GRanges object with editing spacers, three prime extensions, and nicking spacers. The three prime extensions consist of a reverse transcription template (which contains the requested edit) and primer binding site and are to be cloned into the gRNA plasmid as a single entity. Spacer ranges, sequences, off-target counts, and on-targets scores are provided for both editing and nicking spacers, making it easy to select performant combinations for gRNA design.

## Discussion

CRISPR/Cas9 has become an increasingly versatile tool for genome engineering, with a high innovation rate, leading to a fast emergence of new applications. The very first task in conducting a successful CRISPR-based experiment is a proper gRNA design, choosing efficient gRNA spacers with minimal off- and maximal on-target activity. Although experimentally validated gRNA libraries such as the Brunello library (targeting exons of the human genome) exist for the coding genomes of model organisms, non-model organisms and noncoding genomes lack such accessible resources.

For these and other cases with the need for custom-designed spacers, gRNA design tools become very useful. We reviewed the seven most popular scriptable gRNA tools and compared them to our new generic tool multicrispr. As a result, we found that the increasing number of CRISPR/Cas9-based applications and the trend toward large-scale screening require a new generation of gRNA design tools able to process both custom-defined targets and large numbers of them. We benchmarked all scriptable on/off-target analysis–performing gRNA tools from the Torres-Perez et al (2019) CRISPR/Cas9 review table (with the exception of two, which we did not manage to install) and found none of them to be able to handle the dimensions required when, for example, targeting close to all human genes as represented by the Brunello library (Doench et al, 2016). By contrast, our tool multicrispr, designed for performance and generic usage, scales well to very large datasets. For instance, on a set of 1,723 mouse exons, multicrispr completed an order of magnitude faster (17 min) than popular alternatives such as CHOPCHOP (10 h) and CRISPOR (5 h). The only tool that scales very well toward large target sets is FlashFry. However, it misses an up-to-date on-target scoring algorithm and other functionality needed for the full gRNA design workflow.

Mass targeting with CRISPR/Cas9 libraries has also created interesting niche applications such as the parallel targeting of an overlapping target set of related sequences distributed in the genome (e.g., TFBSs). Such applications intriguingly turn conventions upside down: off-targets are no longer always off-targets, if they occur in the target set and may even be desirable, allowing for the use of a smaller gRNA set. In this context, we defined the term TSSF and implemented it and all further functionality needed to provide high-quality gRNAs for such applications in multicrispr.

Aside from performance and multi-target–related tasks, many novel applications require flexible genome arithmetic functionality before spacer search. Although some applications necessitate target extension, others need flanking, inverse targeting, or explicit avoidance of targets. To be able to handle each of these cases, a flexible, intuitive genome arithmetic functionality is required. Multicrispr provides this functionality combined with an easy-to-use and intuitive "grammar" inspired by the tidyverse paradigm of functional programming (Wickham, 2019). In addition, multicrispr is intended to support the process of gRNA design by visual output functionality that intuitively documents each individual analysis step.

Along this line, multicrispr is the only performant and complete solution to design gRNAs for prime editing, returning prime editing spacers, 3′ extensions, and nicking spacers, ready to clone into PE2 and PE3 systems. In addition, multicrispr takes care of CRISPR applications with specific needs for precision by providing functionality to switch between performant Bowtie1 and precise Aho–Corasick algorithms for off-target analysis.

Driven by its unique functionality and performance, multicrispr paves the way for future custom resources. Examples that come to mind, but are beyond the scope of this introduction of the tool, are the application to annotated molecule subclasses such as long non-coding RNA or miRNA in a genome-wide context. Other examples include the application of prime editing functionality to all 70,000 ClinVar (Landrum et al, 2014) sites to investigate how many of them may per se be targeted/altered. A further straightforward application may be the generation of a global resource of gRNA pairs for the excision of complete exons, by making use of the flanking functionality of multicrispr for annotated exons as a target set and effectively targeting the introns.

In summary, multicrispr defines new standards for gRNA design in terms of performance, modularity, and universality. It supports a plethora of CRISPR/Cas9-based applications, including recent developments with the need for TSSF functionality.

# Materials and Methods

## Speed comparison of tools

For performance tests, we used exon sequences of chr1 of mouse mm10 assembly. Starting with 15,000 bp, seven sets of exon sequences were created randomly, each with approximately twice the total length of the previous one in base pairs. Resulting test sets were saved as FASTA and BED files, respectively.

Each tool was installed according to its documentation in identical conda (Anaconda Development Team, 2016) testing environments. Test sets were processed 10 times with each tool, and resulting values were summarized to control for variances in computing performance. The tools were parameterized as indicated in Table S1.

## Brunello library validation

The Brunello library (Doench et al, 2016) represents a validated set of gRNAs comprising 76,441 spacers targeting 19,114 transcripts (each transcript residing in a different gene and each being targeted by four different gRNAs). We downloaded the Brunello gRNA set descriptions from Addgene (https://www.addgene.org/static/cms/filer_public/8b/4c/8b4c89d9-eac1-44b2-bb2f-8fea95672705/broadgpp-brunello-library-contents.txt), which provides an RefSeq mRNA identifier, cut site position, and spacer orientation for each gRNA in the set. After excluding the positive controls, RefSeq mRNA were mapped to chromosomes and (unique) strands using biomaRt (Ensembl 99, *Homo sapiens*). Next, we were able to extend cut sites to full spacer ranges for 75,232 spacers/18,810 transcripts using a [−17, +2] extension for "+" spacers and a [−16, +3] extension for "−" spacers. Subsequently, we extended each spacer to the (first) smallest, fully enclosing exon (i.e., both spacer and PAM are contained in the exon), using Ensembl 99 exon models, as provided through Bioconductor's AnnotationHub record AH78783. This was successful for 66,696 Brunello spacers/exons in 18,800 transcripts. The resulting set was used for multicrispr validation. Table S2 details the number of spacers/transcripts targeted as a result of the reconstruction sequence.

## Installation

Multicrispr can be installed from within an R environment with:

install.packages("BiocManager")

```
BiocManager::install(version='devel')
BiocManager::install("multicrispr")
```

The python package azimuth (for Doench2016 scoring) can be installed from wihtin R with:

```
install.packages('reticulate')
reticulate::conda_create('azienv', c('python=2.7'))
reticulate::use_condaenv('azienv')
reticulate::py_install(c('azimuth', 'scikit-learn==0.17.1'), 'azienv', pip=TRUE)
```

A Bowtie-indexed BSgenome (required for off-target analysis) can be created with index_genome as shown in the following text. This function needs to be run only once for any particular BSgenome. For the frequently used cases, we created pre-built indices, which are downloaded automatically. Beside the 28 organisms for which Bioconductor provides BSgenomes, another set of 224 organisms in twoBit format through their AnnotationHub interface (Morgan, 2019) is available. These can be converted to a BSgenome (Pages, 2020) and then analyzed with multicrispr.

```
BiocManager::install('BSgenome.Mmusculus.UCSC.mm10')
BiocManager::install('BSgenome.Hsapiens.UCSC.hg38')
index_genome(BSgenome.Mmusculus.UCSC.mm10::
BSgenome.Mmusculus.UCSC.mm10)
index_genome(BSgenome.Hsapiens.UCSC.hg38::
BSgenome.Hsapiens.UCSC.hg38)
```

### Visualization

Graphs were generated via R. Illustrations in the context of genomic positions are generated via the multicrispr plot_intervals function, which can also be called explicitly. It operates on a GRange objects of any length.

```
bsgenome <- BSgenome.Mmusculus.UCSC.mm10::BSgenome.Mmusculus.
UCSC.mm10
bedfile <- system.file('extdata/SRF.bed', package = 'multicrispr')
targets <- bed_to_granges(bedfile, 'mm10', plot = FALSE)
plot_intervals(targets)
```

### Code and data availability

Bioconductor: [https://bioconductor.org/packages/devel/bioc/html/multicrispr.html](https://bioconductor.org/packages/devel/bioc/html/multicrispr.html).
Gitlab: [https://gitlab.gwdg.de/loosolab/software/multicrispr](https://gitlab.gwdg.de/loosolab/software/multicrispr).
Website tool and manual: [https://loosolab.pages.gwdg.de/software/multicrispr/index.html](https://loosolab.pages.gwdg.de/software/multicrispr/index.html).
Pre-calculated indices: [https://s3.mpi-bn.mpg.de/minio/data-multicrispr-2020/](https://s3.mpi-bn.mpg.de/minio/data-multicrispr-2020/).

## Supplementary Information

## Acknowledgements

This work was funded by the Deutsches Zentrum für Herz-und Kreislauf-forschung (DZHK, Rhein-Main Site), the Max Planck Society (MPG), and the Cardiopulmonary Institute (CPI).

### Author Contributions

AM Bhagwat: software, formal analysis, and writing—original draft.
J Graumann: conceptualization and writing—review and editing.
R Wiegandt: resources and formal analysis.
M Bentsen: resources and formal analysis.
J Welker: formal analysis.
C Kuenne: resources and formal analysis.
J Preussner: resources and formal analysis.
T Braun: funding acquisition and writing—review and editing.
M Looso: conceptualization, supervision, funding acquisition, investigation, visualization, methodology, project administration, and writing—original draft, review, and editing.

### Conflict of Interest Statement

The authors declare that they have no conflict of interest.

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
