## [Reviewer comments · Life Science Alliance]

Life Science Alliance

multicrispr: fast gRNA designer enables prime editing and parallel targeting of thousands of targets

Aditya Bhagwat, Johannes Graumann, Rene Wiegandt, Mette Bentsen, Jordan Welker, Carsten Kuenne, Jens Preussner, Thomas Braun, and Mario Looso

DOI: 10.26508/lsa.202000757

Corresponding author(s): Mario Looso, Max-Planck-Institute for Heart and Lung Research

Review Timeline:

Submission Date:	2020-04-29
Editorial Decision:	2020-06-05
Revision Received:	2020-08-17
Editorial Decision:	2020-08-21
Revision Received:	2020-08-27
Accepted:	2020-08-31

Scientific Editor: Shachi Bhatt

Transaction Report:

June 5, 2020

Re: Life Science Alliance manuscript #LSA-2020-00757-T

Dr Mario Looso
Max-Planck-Institute for Heart and Lung Research
Dept.1
Ludwigstraße 43
Bad Nauheim, Hessen 61231
Germany

Dear Dr. Looso,

Thank you for submitting your manuscript entitled "multicrispr: fast gRNA designer enables prime editing and parallel targeting of thousands of targets" to Life Science Alliance. The manuscript was assessed by expert reviewers, whose comments are appended to this letter.

The points raised by the reviewers should be addressed. A particular focus should be to ensure the usability of the package - thus I would see a publication of the package in Bioconductor as a prerequisite for publication. This would address the issues raised by reviewer 2 regarding installation and dependencies of the package. I agree that the website proposed by reviewer 2 will be an addition, however I don't see this as a requirement for acceptance since R/Bioconductor is a well used tool even in experimental lab. It may however increase the reach of the tool.

Thank you for this interesting contribution to Life Science Alliance. We are looking forward to receiving your revised manuscript.

Sincerely,

Reilly Lorenz
Editorial Office Life Science Alliance
Meyershofstr. 1
69117 Heidelberg, Germany
t +49 6221 8891 414
e contact@life-science-alliance.org
www.life-science-alliance.org

B. MANUSCRIPT ORGANIZATION AND FORMATTING:

Reviewer #1 (Comments to the Authors (Required)):

In this paper, the Authors introduce the multicrispr package. As they point out, over 100 computational tools for various aspects of CRISPR/Cas experiments already exist, and single out 6 perhaps most used ones, but argue that the latest developments are often not yet reflected in

their capabilities. The core contributions are to 1) put forth a usable software tool 2) support designs for the recently proposed prime editing approach 3) introduce generic genome arithmetics as a way to operate on sequences.

In general, the paper is pleasantly written, the figures are clean and informative, and the software works as advertised. Whether it is taken up by the field, and thus has lots of impact, remains to be seen, but there are few grounds to reject a useful tool in general. The main value to this reviewer is in the usability, with a nifty feature of excluding off-targets in specific target sets. The speed and genome arithmetics are nice to have, but are not really a major hurdle in practice. I only have minor comments.

- crispr or Crispr => CRISPR
- *S. pyogenes* italicized
- Figure 2E x-axis labels not clear (what are the two numbers - only one described in caption?).

Reviewer #2 (Comments to the Authors (Required)):

Bhagwat et al have designed a novel R package for guide RNA design in experiments using the CRISPR/Cas9 system.

As stated in the introduction, there are already many tools available for guide RNA selection and the authors have sought to include two original features.

1) Rather than starting from a sequence provided by the user, the tool takes genomic positions as input and automatically retrieves flanking genomic sequence to look for guide RNAs that are suitably positioned for the application of interest. Several rules defining the sequence to be screened are already provided but it is also possible to design your own.

2) When designing large sets of guide RNAs, guide RNAs that have off-targets with two or less mismatches are normally excluded to prevent unwanted targeting. However, in specific applications, off-target sequences may also be targets of interest and the tool is designed to keep the corresponding guides in such cases.

Features 1 and 2 are straightforward to implement when using existing software for guide RNA design. Nevertheless, they are interesting features and multicrispr would be a nice addition to existing software if it was made easier to use (reducing dependencies for local installation or making it available through a dedicated website).

Major issues

1. A website version of the software would be a useful addition. This would make the tool accessible to many more users.
2. I asked for help from a colleague who is a computer scientist with expertise in bioinformatics and guide RNA design. Despite trying several different ways (see below), he was not able to install the software and check its basic functions.
3. The authors show that their tool is faster than other commonly used tools but it is not clear why. For instance, multicrispr is found to be much faster than CHOPCHOP but, in fact, both use the same bowtie algorithm to look for off-targets and would therefore not be expected to be so different.
4. At least, one guide RNA design tool, Flashfry, has been optimized for runtime. It would be useful to include it in the benchmarking.

5. In order to be useful for design of guide RNAs for prime editing, the software should also check for the presence of suitable PE3 guide RNAs needed to introduce a nick on the opposite strand. The software should also include automatic design of pegRNAs (including PBS and RT template matching user defined inputs) in order to be useful. This is essential for large scale design of pegRNAs.

Minor comments

1. An important issue with off-target analysis is precision. Many tools, in particular tools using bowtie, miss off-targets when scanning the target genome. Some analysis should be done to examine how multicrispr performs with respect to this basic feature of guide RNA design software.
2. Throughout the manuscript, correct « Crispr/Cas9" to "CRISPR/Cas9"
3. Even though they are called guide RNAs, guide RNAs do not actually "guide" Cas9 to the target sequence. In short, Cas9 screens the genome for PAM motifs and unwinds upstream DNA. If the spacer can anneal to the upstream sequence, DNA cleavage takes place.

Page 1 Abstract: "and filters gRNAs on both efficiency and specificity, ». Correct to and "filters gRNAs on both efficiency and specificity predictions ».

Page 1 Abstract: "render multicrispr the tool of choice". Change to "render multicrispr a tool of choice"

Page 1 Correct to "Crispr loci, first reported in 1987"

Page 1 Correct to "When followed by an NGG protospacer adjacent motif (PAM), a genomic sequence identical to the spacer is cleaved by the Cas9 nuclease" or "When Cas9 binds a NGG motif, DNA will be cleaved if the sequence immediately upstream is identical to the spacer sequence"

Page 1 Change to "extended guide RNA consisting of a spacer, the scaffold, a primer binding site, and a reverse transcription template."

Page 1 Correct "While the spacer continues to guide the complex to cut a specific genomic locus, »

Error messages

Using RStudio from lab server or after download onto laptop both failed:

Error: Failed to install 'unknown package' from Git:

(converted from warning) dependency 'latticeExtra' is not available

Error: Failed to install 'unknown package' from Git:

there is no package called 'BiocManager'

And finally after forcing BiocManager install :

g++: error: unrecognized command line option '-fno-plt'

g++: error: unrecognized command line option '-fno-plt'

make: *** [Makefile:232: bowtie-build-s] Error 1

ERROR: compilation failed for package 'Rbowtie'

* removing '/cluster/home/max/miniconda3/envs/r-env/lib/R/library/Rbowtie'

Error: Failed to install 'unknown package' from Git:

(converted from warning) installation of package 'Rbowtie' had non-zero exit status

Reviewer #1

1. In this paper, the Authors introduce the multicrispr package. As they point out, over 100 computational tools for various aspects of CRISPR/Cas experiments already exist, and single out 6 perhaps most used ones, but argue that the latest developments are often not yet reflected in their capabilities. The core contributions are to 1) put forth a usable software tool 2) support designs for the recently proposed prime editing approach 3) introduce generic genome arithmetics as a way to operate on sequences. In general, the paper is pleasantly written, the figures are clean and informative, and the software works as advertised. Whether it is taken up by the field, and thus has lots of impact, remains to be seen, but there are few grounds to reject a useful tool in general. The main value to this reviewer is in the usability, with a nifty feature of excluding off-targets in specific target sets. The speed and genome arithmetics are nice to have, but are not really a major hurdle in practice.

We thank the reviewer for this positive feedback regarding our tool.

2. I only have minor comments.

- crispr or Crispr => CRISPR
- *S. pyogenes* italicized
- Figure 2E x-axis labels not clear (what are the two numbers - only one described in caption?).

We apologize for the inconsistency in using different versions of writing CRISPR and changed it accordingly. We changed Fig2E in order to improve clarity. Now it is given as: "... (E) *Runtime comparison of gRNA design tools: the X-axis depicts increasing number of input sequences and total bases respectively, while the Y-axis shows the total time needed by individual tools to design respective gRNAs on a log10 scale in seconds. ...*"

Reviewer #2

Bhagwat et al have designed a novel R package for guide RNA design in experiments using the CRISPR/Cas9 system. As stated in the introduction, there are already many tools available for guide RNA selection and the authors have sought to include two original features. 1) Rather than starting from a sequence provided by the user, the tool takes genomic positions as input and automatically retrieves flanking genomic sequence to look for guide RNAs that are suitably positioned for the application of interest. Several rules defining the sequence to be screened are already provided but it is also possible to design your own. When designing large sets of guide RNAs, guide RNAs that have off-targets with two or less mismatches are normally excluded to prevent unwanted targeting. However, in specific applications, off-target sequences may also be targets of interest and the tool is designed to keep the corresponding guides in such cases. Features 1 and 2 are straightforward to

implement when using existing software for guide RNA design. Nevertheless, they are interesting features and multicrispr would be a nice addition to existing software if it was made easier to use (reducing dependencies for local installation or making it available through a dedicated website).

We thank the reviewer for this positive assessment of our work.

1. A website version of the software would be a useful addition. This would make the tool accessible to many more users.

We appreciate the concerns of the reviewer regarding the accessibility of multicrispr and the idea of a web version of the tool. However, multicrispr is intended as a generic and flexible tool with a plethora of options and parameters, and designed to be used from within other chunks of code. These features make it rather hard to design an intuitive and user friendly web frontend. A more focused approach on a subset of multicrispr functionality might be more expedient, as we are planning to design and establish a website to search for paired gRNAs for CRISPR based excision applications. However, this project is ongoing and beyond the scope of this manuscript, where we want to introduce multicrispr to the community as a generic tool, able to cope with a multitude of gRNA design tasks.

2. I asked for help from a colleague who is a computer scientist with expertise in bioinformatics and guide RNA design. Despite trying several different ways (see below), he was not able to install the software and check its basic functions.

We apologize for any inconvenience when installing our R package. Meanwhile, the multicrispr package was accepted at Bioconductor, providing a simplified installation routine, given as:

```
install.packages("BiocManager") # Install BiocManager
BiocManager::install(version='devel') # Install development version of Bioconductor
BiocManager::install("multicrispr") # Install multicrispr
```

3. The authors show that their tool is faster than other commonly used tools but it is not clear why. For instance, multicrispr is found to be much faster than CHOPCHOP but, in fact, both use the same bowtie algorithm to look for off-targets and would therefore not be expected to be so different.

We performed comparative benchmarking with all other tools in the same hardware environment, involving spacer identification and on-target scoring, as well as off-target scoring in order to mimic a complete workflow of gRNA design prior candidate filtering. All runs were performed multiple times in order to correct for technical variance. Therefore, the time ranges reported constitute a mean “complete workflow” time for the individual datasets of different sizes tested. The performance difference between e.g. CRISPRseek and multicrispr can be mainly attributed to the off-target analysis method used (Aho-Corasick algorithm versus Bowtie1). CHOPCHOP, also using Bowtie1, is different to multicrispr in the way how Bowtie1 is utilized. Within

multicrispr, we perform just a single Bowtie1 run and later aggregate the results per hit, while CHOPCHOP involves multiple runs. We now detailed this information within the manuscript in the Benchmarking and feature comparison section.

4. At least, one guide RNA design tool, Flashfry, has been optimized for runtime. It would be useful to include it in the benchmarking.

According to the reviewer's request, we performed an additional comparison of Flashfry and multicrispr and added the results to the manuscript, outlined in Supp Fig. 1 and the benchmark section. We want to point out that we had not included Flashfry originally in the comparison because we had focused on tools that provide Doench2016 on-target scoring, which is the current standard. Flashfry instead uses the outdated Doench2014 method for on-target scoring. Multicrispr supports the switch to Doench2014. For the comparison, we utilized a large chr1 subset. Flashfry has also been included into the feature table 2 and in the overview Figure 1C. As illustrated in the new supp figure, we conclude the performance of flashfry to be comparable to multicrispr.

5. In order to be useful for design of guide RNAs for prime editing, the software should also check for the presence of suitable PE3 guide RNAs needed to introduce a nick on the opposite strand. The software should also include automatic design of pegRNAs (including PBS and RT template matching user defined inputs) in order to be useful. This is essential for large scale design of pegRNAs.

We agree with the reviewer on the importance of PE3 guides and implemented this functionality, now available in the latest version of our bioconductor package. It now contains full support for both PE2 and PE3 systems. It returns prime editing spacers, 3' extensions (with RT template, which contains the desired edits, as well as primer binding site), and nicking spacers, the three components that require cloning into a gRNA plasmid. Fig1 has been updated to illustrate the nicking spacer, 40-90 NT downstream of the prime editing cut site, and on the opposite strand. Fig 3 has also been updated: it now includes nicking spacers in the graphical output as well as the returned GRanges.

6. An important issue with off-target analysis is precision. Many tools, in particular tools using bowtie, miss off-targets when scanning the target genome. Some analysis should be done to examine how multicrispr performs with respect to this basic feature of guide RNA design software.

We agree that successful off-target analysis requires high precision. To better understand the precision of Bowtie1, we compared Bowtie's off-target analysis results with those obtained by the Aho-Corasick string matching algorithm, which guarantees precision through exact string matching, and added the results to the benchmark section. Aho-Corasick is much slower (hours versus seconds). By Performing off-target analysis for the four main Prime Editing targets of Anzalone et al. (2019), added to the manuscript as new table 1, we found Bowtie to achieve perfect precision when it comes to exact-match and single-mismatch off-targets. Yet, for double mismatches some off-targets are missed, and for triple mismatches a larger number of off-targets are missed. Following these results, we conclude to use Bowtie1 for large- and medium-scale applications, and if run-time is not an issue, the Aho-Corasick algorithm. To allow for this, the multicrispr now contains an additional

argument *offtargetmethod*, which allows switching between the two methods bowtie or Aho-Corasick.

7. Throughout the manuscript, correct « Crispr/Cas9" to "CRISPR/Cas9"

We apologize for the inconsistency and changed the writing of CRISPR accordingly.

8. Even though they are called guide RNAs, guide RNAs do not actually "guide" Cas9 to the target sequence. In short, Cas9 screens the genome for PAM motifs and unwinds upstream DNA. If the spacer can anneal to the upstream sequence, DNA cleavage takes place. Page 1 Abstract: "and filters gRNAs on both efficiency and specificity, ». Correct to and "filters gRNAs on both efficiency and specificity predictions ».

Changed as suggested by the reviewer.

Page 1 Abstract: "render multicrispr the tool of choice". Change to "render multicrispr a tool of choice"

Changed as suggested by the reviewer.

Page 1 Correct to "Crispr loci, first reported in 1987"

Changed as suggested by the reviewer.

Page 1 Correct to "When followed by an NGG protospacer adjacent motif (PAM), a genomic sequence identical to the spacer is cleaved by the Cas9 nuclease" or When Cas9 binds a NGG motif, DNA will be cleaved if the sequence immediately upstream is identical to the spacer sequence"

Changed as suggested by the reviewer.

Page 1 Change to "extended guide RNA consisting of a spacer, the scaffold, a primer binding site, and a reverse transcription template."

Changed as suggested by the reviewer.

Page 1 Correct "While the spacer continues to guide the complex to cut a specific genomic

locus, »

Changed as suggested by the reviewer.

9. Error messages

Using RStudio from lab server or after download onto laptop both failed:

Error: Failed to install 'unknown package' from Git:

(converted from warning) dependency 'latticeExtra' is not available

Error: Failed to install 'unknown package' from Git:

there is no package called 'BiocManager'

And finally after forcing BiocManager install :

g++: error: unrecognized command line option '-fno-plt'

g++: error: unrecognized command line option '-fno-plt'

make: * [Makefile:232: bowtie-build-s] Error 1**

ERROR: compilation failed for package 'Rbowtie'

*** removing '/cluster/home/max/miniconda3/envs/r-env/lib/R/library/Rbowtie'**

Error: Failed to install 'unknown package' from Git:

(converted from warning) installation of package 'Rbowtie' had non-zero exit status

We apologize for the insufficient installation instructions. We assume our new instructions given above will work for you. Otherwise, please let us know by opening an issue on the GitHub page (<https://github.com/loosolab/multicrispr>). These types of errors could indicate an outdated software environment (Linux distribution version, R version etc. or other problems with packages in the dependency list that have some restrictions regarding the environment), but it might take some back-and-forth to be solved.

August 21, 2020

RE: Life Science Alliance Manuscript #LSA-2020-00757-TR

Dr. Mario Looso
Max-Planck-Institute for Heart and Lung Research
Dept.1
Ludwigstraße 43
Bad Nauheim, Hessen 61231
Germany

Dear Dr. Looso,

Thank you for submitting your revised manuscript entitled "multicrispr: fast gRNA designer enables prime editing and parallel targeting of thousands of targets". We would be happy to publish your paper in Life Science Alliance pending final revisions necessary to meet our formatting guidelines.

Along with the points listed below, please also address the following in the revision,

- Need main article as .doc
- Each table (Table 1, 2, S1) should be attached as a separate editable file

A. FINAL FILES:

-- Summary blurb (enter in submission system): A short text summarizing in a single sentence the study (max. 200 characters including spaces). This text is used in conjunction with the titles of papers, hence should be informative and complementary to the title. It should describe the context and significance of the findings for a general readership; it should be written in the present tense

and refer to the work in the third person. Author names should not be mentioned.

B. MANUSCRIPT ORGANIZATION AND FORMATTING:

Sincerely,

Shachi Bhatt, Ph.D
Executive Editor
Life Science Alliance
www.life-science-alliance.org

August 31, 2020

RE: Life Science Alliance Manuscript #LSA-2020-00757-TRR

Dr. Mario Looso
Max-Planck-Institute for Heart and Lung Research
Dept.1
Ludwigstraße 43
Bad Nauheim, Hessen 61231
Germany

Dear Dr. Looso,

Thank you for submitting your Research Article entitled "multicrispr: fast gRNA designer enables prime editing and parallel targeting of thousands of targets". It is a pleasure to let you know that your manuscript is now accepted for publication in Life Science Alliance.

DISTRIBUTION OF MATERIALS:

Congratulations on a very nice paper. I hope you found the review process to be constructive and are pleased with how the manuscript was handled editorially. We look forward to future exciting submissions from your lab.

Sincerely,

Shachi Bhatt, Ph.D.,
Executive Editor
Life Science Alliance

e contact@life-science-alliance.org
www.life-science-alliance.org